# Allelopathic Effect of Selected Rice (*Oryza sativa*) Varieties against Barnyard Grass (*Echinochloa cruss-gulli*)

**DOI:** 10.3390/plants10102017

**Published:** 2021-09-26

**Authors:** Ferdoushi Rahaman, Abdul Shukor Juraimi, Mohd Y. Rafii, Md. Kamal Uddin, Lutful Hassan, Abul Kashem Chowdhury, H. M. Khairul Bashar

**Affiliations:** 1Department of Crop Science, Faculty of Agriculture, University Putra Malaysia (UPM), Serdang 43400, Malaysia; gs53678@student.upm.edu.my (F.R.); mrafii@upm.edu.my (M.Y.R.); gs53632@student.upm.edu.my (H.M.K.B.); 2Department of Land Management, University Putra Malaysia (UPM), Serdang 43400, Malaysia; mkuddin@upm.edu.my; 3Department of Genetics and Plant Breeding, Faculty of Agriculture, Bangladesh Agricultural University, Mymensingh 2202, Bangladesh; lutfulhassan@yahoo.co.uk; 4Department of Genetics and Plant Breeding, Faculty of Agriculture, Patuakhali Science and Technology University, Dumki, Patuakhali 8602, Bangladesh; kashem@pstu.ac.bd

**Keywords:** allelopathy, alellochemicals, inhibition, germination, average percent inhibition

## Abstract

Rice has been subjected to a great deal of stress during its brief existence, but it nevertheless ranked first among cereal crops in terms of demand and productivity. Weeds are characterized as one of the major biotic stresses by many researchers. This research aims to determine the most potential allelopathic rice variety among selected rice accessions. For obtaining preeminent varieties, seventeen rice genotypes were collected from Bangladesh and Malaysia. Two prevalent procedures, relay seeding and the sandwich technique were employed to screen the seventeen rice (donor) accessions against barnyard grass (tested plant). In both approaches, only the BR17 variety demonstrated substantial inhibition of germination percentage, root length, and dry matter of barnyard grass. The rice variety BR17 exclusively took the zenith position, and it inhibited the development of barnyard grass by more than 40–41% on an average. BR17 is originated from KN-1B-361-1-8-6-10 (Indonesia) and developed by the Bangladesh Rice Research Institute (BRRI), Gazipur, Bangladesh in 1985, having a high yielding capacity of more than 6 t/ha. Our study suggested that the usage of the allelopathy-weed inverse relationship to treat the weed problem can be a fantastic choice in the twenty-first century.

## 1. Introduction

Rice is one of the most significant irreplaceable staples foods in many countries of the world and its availability is connected to food security [1]. Rice is the seed of the monocot plant *Oryza sativa* (Asian rice) and *Oryza glaberrima*, which belong to the grass family, Poaceae, and has twenty wild species and two cultivated *Oryza sativa* (Asian rice) and *Oryza glaberrima* (African rice) [2]. The growing global population has emphasized increasing food production, particularly rice, which is a staple diet for the majority of the world’s population [3]. Weeds, however, have a tremendous impact on agricultural production and weed losses have exceeded all other agricultural damage instigated by insects, nematodes, disease, rodents, etc [4]. Bangladesh, as well as Malaysia’s self-sufficiency policy, has centered on rice, the country’s primary staple food [5]. Therefore, improving the rice yield is always the priority for rice breeding [6].

Weeds are unwanted plants that do not give farmers economic output that is otherwise difficult for farmers to control [7]. It also prerequisites to be prudently managed to limit the effect of weeds on crop yields [8]. This is particularly concerning because barnyard grass is one of the top 15 weed species capable of herbicide resistance [9], with cases reported in 23 countries, primarily in rice but also grown extensively in agricultural fields [10]. It is a prolific species with high tillering potential [11], apart from that, their biological and ecological similarities to rice make this species one of the world’s most common concerns [12].

Allelopathy is a chemical technique that enables a plant to compete for a limited number of resources [13]. According to Zhijie Zhag [14], allelopathy (chemical interactions between plants) has been shown to affect individual performance, community organisation, and plant invasions. Several compounds have been identified as potential rice allelochemicals, including phenolic acids, fatty acids, phenylalkanoic acids, hydroxaicacids, terpenes, indoles, the diterpenoid momilactones, etc [15]. Cell division and proliferation, cellular morphology, cell surface permeability, antioxidant activity, plant growth regulatory system, photosynthesis, water and nutrient uptake, and other physiological and biochemical processes in plants have shown to be affected by allelochemicals [16]. Allelopathic substances impede plant development at all phases, from seed germination to maturity, including seed sprouting, seedling development, dry matter, and biochemical components, according to several studies [17]. Allelopathy is one of the numerous plant interactions regulated by a variety of elements including nutrition, light, temperature, humidity, and others [18]. Rice’s allelopathy has been thoroughly examined, and a wide variety of rice types have been proven to hamper the development of various plant species during co-cultivation [19]. When looking at the behaviour of target crops, it has been observed that rice root and shoot growth are the most sensitive to the stress imposed by allelochemicals [20]. Extracts from the jungle rice shoot have shown a greater inhibitory effect on root length and seedling dry weight, according to Sitthinoi [21]. Soleymani and shahrajabian [22] found that sesame extract density and plant organs had a significant impact on percent germination, coleoptile weight, radical, and coleoptile length. Kadioglu et al. [23] found that aqueous extracts of different weeds can inhibit the germination process, length of root and shoot, and crops’ dry weight, and Chung et al. [24] stated that water-soluble phytochemicals in rice can prevent the germination of *E. cruss-galli*.

Allelochemicals produced by rice cultivars likely impacted the nitrogen intake of adjacent plants, affecting the growth of shoot length of the tested plants [25]. Allelochemicals generated by one crop species can affect the growth, productivity, and quality of crops belonging to the same crop family [21]. Allelopathy is a trait of herbs and grasses, as well as the main crops, which are normally part of the Poaceae family (Grass family). This promises to be a more effective means of protecting the area in the future [26].

Research on the allelopathic capacity of several rice cultivars against spinach (*Spinacia Oleracea*) was published by [27]. The allelopathic efficacy of some rice accessions versus barnyard grass was also examined by [28,29]. Jafari et al. [30] investigated the allelopathic potential of rice (*Oryza sativa* L.) varieties (*Echinochloa crus-galli*) on barnyard grass. Although allelopathy was found in some rice accessions from Bangladesh and Malaysia against various test plants, still, it is unknown which accessions are the best against barnyard grass. There are no published articles that identify the best allelopathic accession between Bangladesh and Malaysia at this time. Resistant weeds are becoming more common, so biological control with allelopathic rice is important. Herbicide varieties are a useful option for weed control in rice fields. This demands a bigger population for germplasm screening, resulting in very confusing research. If chemical weed control is truly required, allelopathic rice may minimise the herbicidal dosage. There is a surge of interest in exploiting plant allelopathy to improve weed management approaches in agriculture over and above to develop new weed control strategies to overcome the limitations of synthetic herbicides. All of these constraints have been alleviated to a considerable extent by using plant breeding. Increasing rice yield, avoiding the use of chemical pesticides, is alternatively a critical challenge in agricultural production. Therefore, precise weed management is required to ensure global food security. This research would support in lowering dependency on chemical herbicides, ecologically sound as well as resolving food security together with boosting the national economy. Consequently, the goal of this study was to screen and identify the most potentially allelopathic rice varieties along with quantifying the percentage of barnyard grass growth inhibition caused by allelopathic effects of rice accessions.

## 2. Materials and Methods

### 2.1. Bioassays

In laboratory-based donor-receiver bioassays, seventeen rice varieties (including both modern and traditional rice accessions) were used to identify the best allelopathic rice variety for further improvement through breeding. The laboratory bioassays were carried out in the Weed Science Laboratory, Universiti Putra Malaysia. Serdang, Selangor, Malaysia. The most practical and easy approach to measuring rice’s allelopathic potential is by using bioassays [31]. The selected rice varieties (Table 1) were collected from the Bangladesh Rice Research Institute (BRRI, Joydebpur, and Gazipur, Bangladesh), Seed Gene Bank, Universiti Putra Malaysia (UPM), and Malaysian Agricultural Research and Development Institute (MARDI), Malaysia, for the screening purposes (Table 1) and the *E. crusgalli* seeds were obtained at Ladang 15 UPM, Malaysia. Using two bioassay methods, the sandwich and relay procedures, the improved allelopathic rice variety was grown for enhanced excellence.

Several rice varieties/lines were filtered out as allelopathic rice in a preliminary finding using simplified laboratory settings [32]. The experiment was set up in a completely randomized design (CRD). The number of seeds that germinated, the length of the shoots, the length of the roots, and the dry matter weight of the barnyard grass were all recorded. In order to calculate the average percent (API) inhibition, the percentage of seed germination, shoot length, root length, and dry matter weight were assessed.

#### 2.1.1. Sandwich Method

Fuji et al. [33] made known the sandwich technique in Japan, where diverse plant elements such as leaves, stem husk, and smashed seeds are inserted between two layers of agar solution. However, in our study, Fuji’s approach was slightly modified, where 5 mL of 2% (*w*/*v*) agar solution was placed into sterilized tiny plastic pots, which were then left undisturbed to solidify. Afterwards, 45 mg of broken rice with husk from each accession was uniformly placed in each pot on the solidified agar medium. The same amount of agar solution was poured again, and 20 barnyard grass seeds were placed on agar medium in each pot. The control sample was prepared with agar medium only. All Petri dishes were placed in an incubation chamber (25 ± 2 °C) and the experiment was continued for 10 days. Barnyard grass seedlings were trimmed to 15 seedlings per container on the fourth day following seed germination. To assess their dry weight, 10-weed seedlings were oven-dried for three days at 72 °C after ten days. According to the theory, smashed rice generated allelochemicals in the agar media which would regulate seed germination and seedling growth of barnyard grass. The barnyard grass seedlings were trimmed to 15 seedlings per Petri dish after four days of seed germination. Several seedling parameters, such as percent germination, mean germination time (day), root length (mm), shoot length (mm), dry weight (mg), and average percent inhibition (API), were determined for 10 randomly selected barnyard grass seedlings from each Petri dish. The experiment was repeated four times in a completely randomized design (CRD).

#### 2.1.2. Relay Seeding Method

Navarez and Olofdotter [34] first used this laboratory bioassay approach to test the allelopathic potential of rice at the International Rice Research Institute (IRRI), Philippines. Twenty rice seeds from a single rice accession were placed in each Petri dish, lined with 9 cm Whatman No. 1 filter paper, and soaked in distilled water for 10 min. Afterwards, the Petri dish was placed at UPM’s weed science laboratory in Malaysia at room temperature (30 ± 2 °C) and with a 12 h light period. The rice seedlings were then reduced to ten per Petri dish, and twenty barnyard grass seeds were placed as a co-culture together with the 12-day old sprouting rice seedlings. In the control treatment, only 20 seeds of barnyard grass were kept in Petri dishes, and the experiment was continued up to 22 days, i.e., 4-leaf stage. To ensure proper hydration as well as absolute germination and development, 3 mL of distilled water was supplied to each petri dish every two days. Subsequently, the rice and weed seedlings were gently plucked and cleaned thoroughly. In a completely randomized design (CRD), the treatments were replicated four times. Besides, the 10-weed seedlings were oven-dried at 72 °C for three days and their dry weight was measured. The main idea behind the methodology was that rice seedlings emitted allelochemicals in the co-culture, which affected the growth of the test plant (barnyard grass) seedlings.

### 2.2. Data Collection

ImageJ software was used to record the root and shoot lengths [35]. Starting four days after seed placement, the germination percentage was recorded. All of the rice accessions provided data for the 10-seedling attributes of germination (Ger, %), mean germination time (MGT, day), shoot length (cm), root length (cm), dry weight (DW, mg), and average percent inhibition (API, %). The following formula, as described by Nicols and Heydecker [36], was used to calculate the mean germination time (MGT):(1)$$\mathrm{MGT}\;=\;\frac{\Sigma nT}{\Sigma nt}$$
where, n = number of newly germinated seed at ‘T’ time and t = days from sowing

For example, the number of seeds germinated after 4, 6, 8 and 10 were recorded and used to calculate mean germination time. Both methods were used to measure the root and shoot lengths (mm) of five randomly chosen *E. crus-galli* seedlings, 17 days after seed planting. The seedlings were placed in paper bags and heated for 72 h in an electric oven set to 80 °C. To calculate the percent reduction in germination, mean germination time, root length, shoot length, and dry matter accumulation, the following mathematical equation was used Kabir et al. [27]:(2)$$\mathrm{Percent}\;\mathrm{reduction}=\frac{C\;-\;T}{C}\times \;100$$
where C denotes the value under the control treatment (without rice) and T denotes the value under the rice treatment.

Average per cent inhibition was used to analyze the overall impacts (API) of rice varieties on the growth of *E. crus-galli* and it is expressed by Karim et al. [29]:API = (% root reduction + % shoot reduction + % dry weight reduction + % germination reduction + % mean germination time reduction)/5.(3)
where 5 denote the total number of parameters covered. API levels were employed as an indication of rice varieties’ allelopathic potential. A greater API score implies that the cultivars have a higher allelopathic potential and vice versa.

### 2.3. Statistical Analysis

ANOVA procedure and means were separated by the Duncan Multiple Range Test using SAS software (version 9.4). The correlation between API and all seedling traits of barnyard grass were calculated using PROC CORR. The widely used NTSYSpc 2.02e (Numerical Taxonomy and Multivariate Analysis) System was utilized to test various types of agglomerative cluster analysis of some type of similarity or dissimilarity matrix to better understand the similarities between the rice accessions studied [37].

## 3. Results

### 3.1. Germination Percentage

In the sandwich method, the range of germination percentage was 100–76.16%. The allelopathic effects of rice accessions significantly (*p* < 0.01) regulated the germination percentage of barnyard grass. The zenith germination percentage was noticed in the control (100%) and the lowermost was in BR17 (76.16%) and it was going along with Line (16-3-38-9) (80.33%) and BRRI dhan75 (80.83%) (Table 2). The BR17 germination percentage was 23.84% lower than the control. The increased proportion of barnyard grass seed germination in BR26 (100%), BRRIdhan 48, and MRQ 74 (98.33%) was related to the variety’s less suppressive allelopathic impact. On the other hand, rice variety BR17 had the highest percent reduction of barnyard grass seed germination (23.84%), was statistically identical to BR23 (21.21%), and it was near to the BRRIdhan 45 (8.34%). A good number of rice varieties were identified that caused a 19.17–23.84% reduction of germination of barnyard grass.

In the relay seeding method, the extreme germination appeared in the control and it was succeeded by MRQ74 (85.83%), Saitta (85.83%), BRRIdhan 45 (85%), MRIA1 (85%), BRRI dhan75 (85%), and BR17 (73.58%) (Table 2. Again, the percent reduction of germination was detected 11.92–18.9% in 5 varieties and 5.50–9.17% in 12 rice varieties. Among all varieties, BR17 displayed the highest germination reduction (18.9%), and the lowest was seen in Saitta (5.50%) (Table 2).

### 3.2. Mean Germination Time

The mean germination time was extremely influenced by the allelopathic effect. The mean germination fluctuated between 5.63 days to 3.76 days. The control took the crown position where BR17 acquired the lowest. The BR23 and BR26 endorsed the next position, respectively. The results disclosed that a good number of accessions retained the neighboring position to BR17 (Table 3). According to Table 3, a small number of variations (4 accessions) were observed which gave rise to a decrease in the mean germination time of barnyard grass, i.e., 21–33%. The BR17 variety had the largest percent reduction in MGT of barnyard grass seed, at 33.07%, while the other variety, Putra 2, had the lowest reduction (6.84%) in the sandwich method (Table 3).

In the relay seeding method, the notable allelopathic effect was also proclaimed in the mean germination time. Germination disruption is a usual phenomenon of the allelopathic influence that also appeared due to aqueous weed extracts in the rice. The value of the mean germination time was between 5.81–3.82 days. The highest mean germination time was detected in the control, which was followed by BRRIdhan45, MRQ74, Saitta, Putra2, and BRRIdhan48. The lowest mean germination time was observed in BR17 and was near to the BR23, Line (16-3-38-9), and BRRIdhan 39 (Table 3). Contrarily, the percentage mean germination time reduction was acquainted 15.33–29.12% from 6 varieties and 9.51–13.88% observed from other varieties. Overall, BR17 showed the highest mean germination time reduction (29.12%) and the lowest from MRQ74 (9.51%) (Table 3).

### 3.3. Root Length

In this study, the influence of crushed seeds of several rice accessions on root length was demonstrated by using the sandwich method. The range of root length was between 1.59 to 0.66. On the basis of root length, the control showed the utmost root length and BR17 presented the minimum root length (cm) compared to the other varieties (Table 4). The control was 0.93% higher than BR17, the treatment BR23 and BRRIdhan 46, BRRIdhan 39 are adjacent to BR17. In the case of percent reduction of root length, it was observed in the range between 50–58.15% in 8 varieties, and 16.07–47.12% was in the other 9 varieties (Table 3). The most allopathic variety was BR17 (58.15%) based on the percent reduction in the root length, while the least was BRRIdhan 48 (16.07%). On the basis of root length reduction, BR17 (58.15%) was the most allelopathic variety, while BRRIdhan 48 (16.07%) was the least.

The allelopathic effects of intact rice seedlings were observed within different allelopathic rice accessions. The control took the top place in terms of root length, which was followed by Saitta, BRRI dhan69, and BRRI dhan82. BR17, however, out of seventeen varieties, exhibited a promising effect on root length which is why barnyard grass took the lowest value when treated with BR17 (Table 4). On the other hand, a 43.06–47.58% reduction of root length was observed from 5 rice varieties, and 6.57–38.81% was observed from others varieties in the relay seeding method (Table 4).

### 3.4. Shoot Length

Rice root exudates had a comparable influence on the shoot length of *E. cruss-galli* as was performed on the weed’s root length. The weed’s branch length decreased as a result of the allelopathic action of rice types. Barnyard grass shoot length was 1.64 in BR17, which was the shortest of all rice accessions. The next lowest was in BR23 (1.70 cm), which was followed by BR26 and then BRRI dhan39 (Table 5). Allelochemicals were produced from shattered rice seeds or root exudates, which impeded the test plant’s shoot development. On the other hand, the percent shoot length decrease was found to be 45–51.90% in six kinds and 9–39.39% in 11 others. BR17 exhibited the greatest loss in shoot length (51.90%) and the least (9.96%) in BRRIdhan 48 in the sandwich method (Table 5).

In the relay seeding method, the shoot lengths were between 3.28–1.94 mm. Like other seedling traits of the test plant, the control received the top position in shoot length. Among all the accessions, BR17 took the lowest position (2.02 mm) and the treatment BR26 (1.94 mm), BRRIdhan 46 (2.12 mm), BR23 (2.15 mm), BRRIdhan 45 (2.22 mm), MRIA1 (2.36 mm), BRRIdhan 39 (2.37 mm) were close to the smallest possessor (Table 5). Table 5 shows that the highest percent reduction of shoot length was observed in BR26 (40.61%) and the lowest (16.99%) was observed in Saitta. On the other hand, a percentage reduction (30.09–40.61%) was observed for 6 varieties, which was 16.99–27.86% for the other 11 varieties (Table 5). Among all the parameters, shoot length was non-significant (Table 5) both in the percentage of shoot length and percent reduction of shoot length.

### 3.5. Dry Matter

The allelopathic influence of rice accessions on weed seedling growth (i.e., root and shoot length) was mirrored in weed dry matter output. The test plant, when treated with BR17, contributed the minimum amount of dry matter production (4.75 mg) and followed by line 16-3-38-9 (5.50 mg) and Putra 1 (5.75 mg), respectively. The dry matter production range of the test plant was between 7.5–4.75 (Table 6). In the table sandwich method, it was observed that dry matter percent reduction was highest in BR17 (36.16%) and lowest in Saitta (3.23%).

The allelochemicals produced by intact rice seedlings also influenced the dry-matter production of the tested plant species (Table 6). In the relay seeding method, the highest dry-matter production was in the control and the lowest was in BR17 (10.5 mg) followed by BRRIdhan 46 (11 mg), BR23 (14 mg), BRRIdhan39 (14 mg), and BRRIdhan75 (15). On the other hand, the highest percentage of dry-matter reduction was in BR23 (75.35%), which was statistically similar to BR17 (75.23%) and the lowest (43.55%) for MRQ74. Overall, 62.36–75.35% dry-matter reductions were observed in nine rice varieties and the other varieties scored 43.55–57.66% (Table 6).

### 3.6. Average Percent Inhibition (API)

Among the seedling traits, the average percent inhibition is the most potential parameter and a good number of researchers have used this parameter to point out the super allelopathic variety. In the sandwich method, the largest value of API was in BR17 (40.62) (Table 7). When grouping was made based on the average percentage inhibition, it was observed that 2 varieties (BR23 and BR26) exhibited more than 30% inhibition, 10 varieties displayed more than 20% inhibition, 3 varieties exposed more than 10% inhibition, and a single variety unveiled a nearly 5% inhibition. Finally, all the parameters were proclaimed highly statistically significant at the 1% level but solely dry weight is significant at the 5% level.

Among the Seventeen accessions, BR17 alone secured the uppermost place in the relay seeding method. On the basis of average percent inhibition (API), the ranking was BR17 > BR23 > Line (16-3-38-9) > BR26 > BRRIdhan46 > BRRIdhan39 > Putra1 > BRRIdhan45 > BRRIdhan75 > BRRIdhan48 > MRIA1 > Line(L50-38-8) > BRRIdhan82 > Putra2 > MRQ74 > BRRIdhan69 > Saitta (Table 7). Among various parameters, API is the most effective regarding allelopathic confirmation because it is the average of all the parameters. In the relay method, the API ranged between 41.52% and 18.79%. The greatest average inhibition was seen in BR17 which was followed by BR23, and the lowest was in Saitta. Again, the percent reduction in API was also observed highest in BR17 (41.52%) and ranged from 32.21–41.52% in 6 varieties. It was lowest in Saitta (18.79%). Besides this, in the other 11 rice varieties, the value fluctuated from 18.79–29.40% and is presented in Table 7.

## 4. Correlation of Different Quantitative Traits

### 4.1. Sandwich Method

In the sandwich method, germination had not only a positive correlation with root length and dry weight but also a highly significant and positive correlation with shoot length and mean germination time, in addition to API (Table 8). On the other hand, the root length was highly and positively correlated with shoot length and API at a 1% level of probability, but merely a positive correlation was seen with dry weight as well as mean germination time at a 5% level. Shoot length was highly correlated with mean germination and API, but exclusively, with a significant positive correlation with dry weight. Alternatively, dry weight had a positive and significant correlation with mean germination time at a 5% probability level and an exceedingly significant and positive correlation with API. API had high significance along with a positive correlation among all the parameters at a 1% probability level.

### 4.2. Relay Seeding Method

In this approach, germination showed a substantial and positive link with root length, shoot length, and dry weight, but a strong and very significant correlation with mean germination time and API. Root length exhibited a positive and significant correlation with shoot length, dry weight, and mean germination time and was only highly significant along with a positive correlation with API (Table 8). Moreover, shoot length had a significant and positive correlation with dry weight and mean germination time. Likewise, dry weight had a positive and significant correlation with mean germination time. Furthermore, mean germination time displayed a highly significant correlation with API. However, all the parameters pointed out a high and positive correlation with API.

## 5. Comparison of Two Methods on Different Seedling Traits

Figure 1 shows the graphical representation of rice accessions based on API, where existed alterations in the parameter during the period of seedling growth when treated with two methods viz. sandwich and relay. The interaction between rice accessions and API was significant. Variations in the graphical depiction show that there are considerable variances amongst rice accessions. Among the rice accessions, BR17 reached the highpoint (more than 40%) and BR23 and BR26 took position nearest to the BR17 (Figure 1). The accessions with the highest API value have strong allelopathic potential. The figure also shows that Saitta was the lowest achiever (below 24%). The inhibition was stronger in the relay seeding method than that of the sandwich method. In both bioassays, the analysis of variance unveiled that all the rice accessions greatly inhibited the different seedling traits of barnyard grass. Previously, many researchers used these two methods and found that it was effective in this research [27,28]. Eventually, the average inhibition percent of BR17 and Saitta had the maximum value gaps.

## 6. Clustering

The dendrogram revealed 17 rice accessions clustered into six groups with a similarity coefficient of 0.90, and this implied a high level of agro-morphological variation of rice accessions (Figure 2). The UPGMA dendrogram generated by cluster analysis indicated the differences in the relationship among the accessions. As a consequence, selecting any of these accessions based on their relationship will improve the breeding program’s results. Cluster I had 5 accessions viz. BRRIdhan 45, BRRIdhan 48, BR17, and Line (16-3-38-9) collected from BRRI, Bangladesh, and UPM, Malaysia. Similarly, cluster II had 5 accessions namely Line (L50-38-8), BR26, BR23, BRRIdhan 39, BRRIdhan 75 from UPM, Malaysia, and BRRI, Bangladesh. Cluster III took 4 accessions viz Saitta, MRQ74, Putra2, BRRIdhan 46 gathered from BRRI, Bangladesh, MARDI, Malaysia, and UPM, Malaysia. Moreover, the cluster IV, V, and VI each took one accession viz MRIA1, BRRIdhan69, and BRRIdhan 82 in turn (Table 9). There are diverse rice accessions domesticated between these ancestors. In comparison to other clusters, the dendrogram demonstrates that variations in one cluster are essentially identical and have less variability. The variations in one cluster shared a lot of the same traits and had little diversity variance. Crossing between varieties within the same group or closely related groups may result in less variance, but crossing between groups far apart will result in greater diversity.

## 7. Principal Component Analysis (PCA)

The principal component analysis (PCA) is the re-validation tool of cluster analysis, as shown in Figure 3 and Figure 4. According to Jobson, PCA is an effective tool [38]. The two-dimensional (Figure 3) and three-dimensional (Figure 4) graphical elucidations revealed that the majority of rice accessions were dispersed at short distances, while five rice accessions were dispersed at long distances, as indicated by the eigenvector. V_1_, V_2_, V_7_, V_11_, and V_12_ were the accessions furthest from the centroid. Other rice accessions were also close to the centroid.

## 8. Discussion

Germination is a consequent enhancement of plant metabolic activity [39]. The yield and quality of any crop plant depend on germination. In the sandwich method, the germination of barnyard grass is low when treated with allelopathic rice accessions, and these actions might have been due to the effect of the allelochemicals released from the rice variety which affected the germination as well as the seedling growth of barnyard grass. Significant allelopathic effects of rice accessions were also observed in the sandwich method on the percentage of germination [28]. Kumbhar and Patel [40] unveiled that the phenolic compounds commonly released from weeds, such as alkaloid, terpenoid, flavonoid, tannin, saponin, influence the germination and establishment of crop stands by disrupting various essential processes such as respiration and enzyme activity. In addition, Kabir et al. [27] showed that due to high phytotoxic effects, the spinach germination percentage decreased in consequence of the rice variety WITA12 and BRRIdhan 44. Ma et al. [41], and other researchers, reported the inhibition of target test seed germination by allelopathic effects of rice accessions. A delay in seed germination was reported by Kruse et al. [42]. Allelochemicals (released from allelopathic rice species) are comparable to plant hormones as it causes delayed germination [43].

The outcome of the relay seeding showed that the allelochemicals released from different rice root exudates impede the seed germination of the tested plant. Different sorts of responses were seen during the relay seeding process when several allelochemicals were produced from the rice seeds spouting, influencing the germination of spinach seeds [27]. All weed species had the greatest germination rates in control treatments compared to all other treatments, suggesting rice accessions had a suppressive impact or allelopathic capacity on weed species germination [44].

For the mean germination time (sandwich), it was observed that some varieties offered low values. The low mean germination time may be due to the effect of p-coumaric acid present in the allelopathic accessions. The rice plants released different secondary metabolites which subsequently inhibited or stimulated the growth and development of the test plant. The total germination of all weed species was likewise pointedly (*p* ≤ 0.01) affected by the ordered aqueous extracts of different rice cultivars [44]. Rice released some allelochemicals, among them, p-coumaric acid has reported being active against barnyard grass at high concentrations [45]. The mean germination time (relay seeding method) findings of this study matched those of Jabeen and Ahmed [46], who studied the effects of *Asphodelus tenuifolius* and *Fumaria indica* on maize seeds and reported that the species were responsible for inhibiting germination. Rice has a comparable inhibitory effect on weed seed germination [47,48].

In the sandwich method, the negative effect of rice exudates caused stunted roots with pruned root tips. Similar findings for *Ipomoea, Cymbopogon, Hyptis, and Alternanthera* were also reported in wheat [49]. Ismail et al. [32] reported that several rice species, particularly Manik and Makmur, reduced *E. crusgalli*’s root length by more than 75% because of the allelopathic impact of rice species. This was consistent with the observations reported by Navarez and Olofsdotter [34]. Momilactones and phenolic compounds are the most common allelochemicals found in rice, according to Jabran [50]. Momilactone A and B, diterpenes originally isolated from rice husks, have been identified as major allelochemicals in rice [51]. Smashed seeds (used this method) contain rice husk that may have played a role in the reduction in germination and growth of roots and shoots.

On the other hand, in relay seeding, it might be due to the phytotoxic effect of the plant. Phenolics present in rice plants play a role in rice allelopathy against weeds [52]. Berendji et al. [53] found that root exudates of allelopathic rice accessions showed the highest inhibitory activity in some traits of barnyard grass seedlings. Furthermore, Kabir et al. [27] discovered that the rice cultivar WITA-12’s allelopathic effect was reduced due to the root length of spinach, i.e., about 60%. In another study, Ismail et al. [32] found that the rice variety Manik and Makmuer could reduce the root length of *E. crus-galli* by more than 80% and 75%, respectively. Khang et al. [54] reported that most of the vanillin and vanillic acid treatments had strong inhibitory effects on lettuce and radish seedling development, as well as high stimulatory activity on rice root elongation.

According to Karim et al. [25], the allelopathic effect (sandwich method) of several rice cultivars developed by BRRI, Bangladesh, caused the greatest shoot length drop (37.18%) in lettuce. Kabir et al. [27] and Karim et al. [25] also found a similar response of rice varieties on the shoot growth of weeds. All of the effects of rice varieties on the shoot length of the test plant give the idea that rice accessions had distinguishing genotypic traits that are linked to allelopathy. The rice cultivars had no stimulating impact on this investigation.

The allelopathic effect (relay method) of diverse rice cultivars developed by BRRI, Bangladesh, resulted in the largest loss of shoot length in lettuce (37.18%) [25]. Allelochemicals can alter the concentrations of plant growth regulators or promote phytohormone imbalances, limiting the growth and development of target plants [28]. Additionally, the concentration of momilactone B in plants and root exudates is greatly raised when rice plants are cultivated near barnyard grasses [55].

For dry matter (sandwich), according to Karim et al. [29], allelopathic rice accession resulted in poor root growth of the test plant due to the allelochemicals produced by the rice plant. In other findings, the allelopathic variety, WITA12, produced the highest drop of dry matter and that was in Kataribhog (79%) [27]. The inhibition rate based on the dry weight of the seedling was higher than that of the emergence and seedling length of barnyard grass [56]. The allelopathic effects of root exudates significantly inhibited the height, number of tillers, and total dry weight of barnyard grass plants [57].

In relay, it might be due to the effect of stunted root length and shoot length. Poor seed germination, effects on the root, shoot development, reduction in dry weight, and the impediment on coleoptiles elongation are the most widely reported morphological effects [29]. In several studies, allelopathic substances have been proven to delay plant development at all stages viz. from seed to maturity, including seed germination, seedling development, leaf area, dry matter, and biochemical components [58]. The greatest inhibitory impact on barnyard grass seed germination, seedling length, and dry weight was found in a bioassay in view of nine recognized allelochemicals and their combinations [59]. Another research study disclosed that allelochemicals resulted in a considerable reduction in root and shoot length, fresh and dry weights, and total chlorophyll and protein content [60].

In sandwich, most of the allelopathic rice accessions displayed outstanding germination inhibition in barnyard grass seed in the sandwich method. The accessions were sorted according to API i.e., BR17 > BR23 > BR26 > Line (16-3-38-9) > Putra1 > BRRIdhan39 > BRRIdhan75 > BRRIdhan46 > Line (L50-38-8)> MRQ74 > Putra2 > Saitta > BRRIdhan69 > BRRIdhan82 > BRRIdhan45 > MRIA1 > BRRIdhan48 (Table 3). Again, the percent reduction of API was highest in BR17 (40.62%), next to BR23 (34.14%) and BR26 (31.07%), and lowest in BRRIdhan 48 (5.81%) (Table 3). It might be due to the phytotoxic effect of all traits. A good number of researchers used this parameter in the detection of allelopathy [29]. According to API, BR17 has placed at the top, followed by BR 23, BR26, and Line (16-3-38-9). Salam et al. [61] suggested that BR17 is the most allelopathic among 102 Bangladeshi rice cultivars. S M R Karim et al. [25] stated that the allelopathic potential of most of the selected rice varieties is good, which resulted in 30 to 35% API, which was supported by Ismail et al. [32]. Mazid et al. [28] observed that rice accession MR73 caused more than 40% API other than some traditional and modern rice accessions also presented > 30% API. Siyar et al. [62] reported that allelochemicals are often associated with cellular damage and abnormalities in the photosystem. So, this is one of the reasons for inhibition.

In the case of the relay seeding method, when many plant species were grown along with rice cultivars in the field and/or laboratory, it was illustrated that the rice cultivars inhibited the growth of the plants [63]. This approach exposed a significant barnyard grass inhibition in all the chosen rice accessions. The changes in germination rate, mean germination time, root length, shoot length, and dry matter amount are all substantially linked with the API.

Weeds raise rice production costs by prompting the use of pesticides to manage them, as well as reducing rice grain quality, lowering net revenue [64]. Molecular genetics (QTL) and biotechnology, as well as traditional breeding procedures, might be employed to generate the selected elite cultivars. Breeding allelopathic rice types would make a significant contribution to long-term rice production [4]. Chemical fertilizers are highly used in modern agriculture, particularly in wheat production [65]. It is critical to address both the economic and ecological benefits of rice field weed management for agricultural development [12]. Due to the deleterious effects of commercial herbicides, it is essential to investigate many alternative weed management strategies [21] and allelopathy promises to be one of them [66]. Allelopathy is an eco-friendly weed management tool, which is practiced to combat the impacts of environmental pollution [67]. Kato-Noguchi and Ino evaluated allelopathic potentiality of 60 conventional and 42 high yielding cultivars, where they assessed against common weeds such as cress, lettuce, *Echinochloa cruss-gralli*, and *Echinochloa colonum*, with Bangladeshi modern rice variety BR17, and displayed the highest allelopathic potentiality across all bioassays [68]. In another research, Kato-Noguchi et al. [69] found a similar finding when they analyzed the allelopathic activity of 102 contemporary and traditional varieties and disclosed that BR17 (modern variety) is the most allelopathic among all. However, donor plants produce allelochemicals that impact receiver plants, and these plants respond to the donor plants by modifying gene expression [67]. In recent years, weed management programs have tended to emphasize nonchemical weed management, i.e., safety measures, or are “environmental or eco-friendly” in general [4]. Rice that possesses allelopathic potential has been reported from all around the world [50].

## 9. Conclusions

In both bioassays, the sandwich method and relay seeding method, a decent amount of allelopathic rice accessions showed excellent inhibitory effects against the barnyard grasses in this investigation. In accordance with the API value, BR17 took a prominent position (40.62%) in the sandwich method, likewise (41.52%) in the relay seeding technique, whereas the BR23, BR26, and Line (16-3-38-9) acquired adjacent values. If a crop’s allelopathic possessions can be improved, it means that the crop’s competitive capacity against weeds can be enhanced, the number of herbicides sprayed can be decreased, and environmental jeopardies can be abolished. Indeed, rice is, without any doubt, one of the world’s most significant cash crops. If allelopathic traits together with high-yielding rice varieties might be ensured, it could be a better option in securing food demand furthermore help to shape an ecofriendly environment in near future. However, further studies will be needed for the identification of allelopathic compounds and their effect in the field.

## Figures and Tables

**Figure 1 plants-10-02017-f001:**
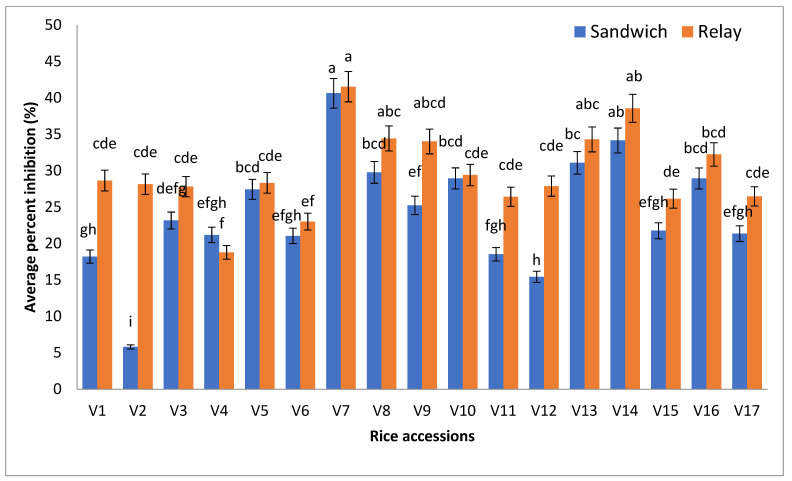
Rice accessions showing average percent inhibition on barnyard grass (*E. crusgalli*) through both the relay and sandwich method. Note: V1 = BRRIdhan 45, V2 = BRRIdhan 48, V3 = Line (L50-38-8), V4 = Saitta, V5 = BRRIdhan 75, V6 = BRRIdhan 69, V7 = BR17, V8 = Line (16-3-38-9), V9 = BRRIdhan 46, V10 = Putra 1, V11 = BRRIdhan 82, V12 = MRIA1, V13 = BR26, V14 = BR23, V15 = MRQ74, V16 = BRRIdhan 39, V17 = Putra 2.

**Figure 2 plants-10-02017-f002:**
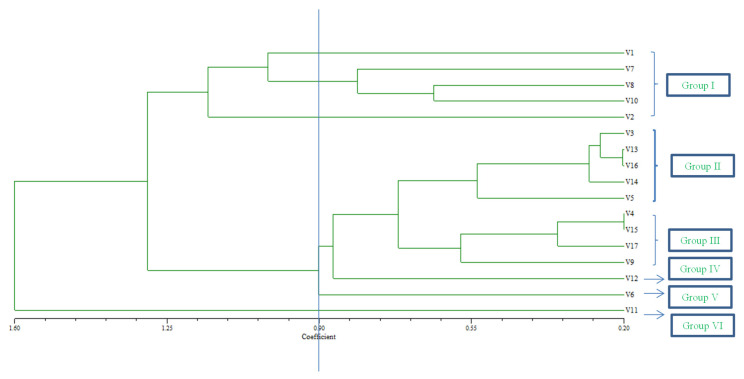
Cluster analysis on the average percent inhibition (API) values of all indicator rice accessions, showing the similarity among indicator rice accessions.

**Figure 3 plants-10-02017-f003:**
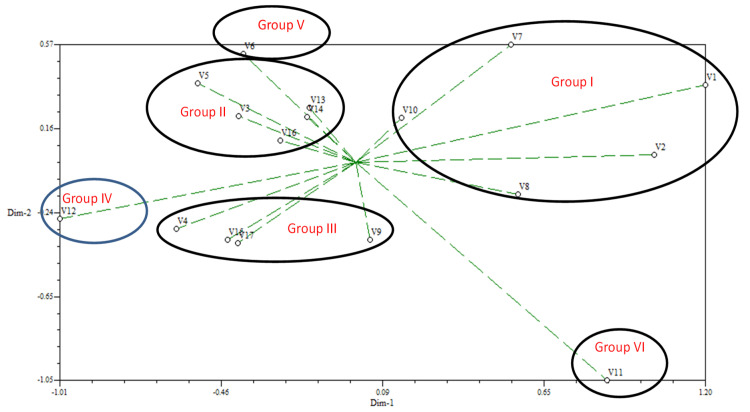
Principal component analysis (PCA)-2D graphical relationship among the rice accessions based on Euclidian distance.

**Figure 4 plants-10-02017-f004:**
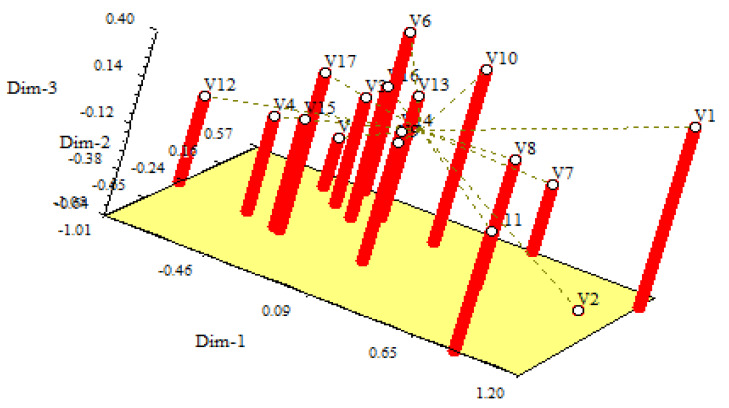
Principal component analysis (PCA)-3D graphical relationship among the rice accessions based on Euclidian distance.

**Table 1 plants-10-02017-t001:** List of different rice accessions used in this study.

Accession Code	Variety	Source	Varietal Speciality	Year of Release
V_1_	BRRIdhan 45	BRRI, Bangladesh	Early maturing	2005
V_2_	BRRIdhan 48	BRRI, Bangladesh	Early maturing	2008
V_3_	Line (L50-38-8)	UPM, Malaysia	Weed competitive and require minimum water	2018
V_4_	Saitta	Local variety, Bangladesh	Traditional	-
V_5_	BRRIdhan 75	BRRI, Bangladesh	Not lodging, Slightly aromatic	2016
V_6_	BRRIdhan 69	BRRI, Bangladesh	Not lodging, late maturing	2014
V_7_	BR17	BRRI, Bangladesh	Suitable for haor (depressed area)	1985
V_8_	Line (16-3-38-9)	UPM, Malaysia	Allelopathic and high zinc content	2018
V_9_	BRRIdhan 46	BRRI, Bangladesh	Photoperiod sensitive, suitable for flood prone areas	2007
V_10_	Putra 1	UPM, Malaysia	High yielding, blast resistant	2021
V_11_	BRRIdhan 82	BRRI, Bangladesh	High amylose content (28%)	2017
V_12_	MRIA1	MARDI, Malaysia	Drought tolerant	2014
V_13_	BR26	BRRI, Bangladesh	Intermediate amylose	1993
V_14_	BR23	BRRI, Bangladesh	Late maturing, photo period sensitive	1988
V_15_	MRQ74	MARDI, Malaysia	High tillering and weed competitive	2005
V_16_	BRRIdhan 39	BRRI, Bangladesh	Early maturing	1999
V_17_	Putra 2	UPM, Malaysia	High yielding, submergence tolerance	2021

Note: BRRI—Bangladesh Rice Research Institute, UPM—Universit Putra Malaysia, MARDI—Malaysian Agricultural Research Development Institute.

**Table 2 plants-10-02017-t002:** The allelopathic outcome of rice accessions on percent germination of barnyard grass employing the sandwich and relay seeding method.

Variety	Code	Sandwich Method		Relay Seeding Method	
		WSG (%)	% Reduction for WSG	WSG (%)	% Reduction for WSG
Control	V_0_	100 ^a^	0	90.83 ^a^	0
BRRIdhan 45	V_1_	91.66 ^b–e^	8.34 ^d–g^	84.99 ^a–c^	6.36 ^b,c^
BRRIdhan 48	V_2_	98.33 ^a,b^	1.69 ^g^	82.49 ^b–d^	9.17 ^b,c^
Line (L50-38-8)	V_3_	87.49 ^d–f^	12.51 ^c–e^	82.49 ^b–d^	9.10 ^b,c^
Saitta	V_4_	90.83 ^c–e^	9.17 ^d–g^	85.83 ^a,b^	5.50 ^c^
BRRIdhan 75	V_5_	80.83 ^f–h^	19.17 ^a–c^	84.99 ^a–c^	6.28 ^b,c^
BRRIdhan 69	V_6_	84.99 ^e–g^	15.00 ^b–d^	83.33 ^b–d^	8.28 ^b,c^
BR17	V_7_	76.16 ^h^	23.84 ^a^	73.58 ^e^	18.9 ^a^
Line (16-3-38-9)	V_8_	80.83 ^f–h^	19.17 ^a–c^	78.82 ^b–e^	13.06 ^a–c^
BRRIdhan 46	V_9_	93.33 ^a–d^	6.663 ^e–g^	83.33 ^b–d^	8.19 ^b,c^
Putra 1	V_10_	87.49 ^d–f^	12.51 ^c–e^	79.99 ^b–e^	11.92 ^a–c^
BRRIdhan 82	V_11_	96.33 ^a–c^	3.67 ^g,f^	84.99 ^a–c^	6.35 ^b,c^
MRIA1	V_12_	91.66 ^b–e^	8.34 ^d–g^	84.99 ^a–c^	6.25 ^b,c^
BR26	V_13_	100 ^a^	15.84 ^b–d^	82.49 ^b–d^	9.06 ^b,c^
BR23	V_14_	91.66 ^b–e^	21.21 ^a,b^	76.82 ^d,e^	15.32 ^a,b^
MRQ74	V_15_	98.33 ^a,b^	9.17 ^d–g^	85.83 ^a,b^	5.46 ^c^
BRRIdhan 39	V_16_	87.49 ^d–f^	11.34 ^d–f^	78.32 ^c–e^	13.88 ^a–c^
Putra 2	V_17_	90.83 ^c–e^	6.67 ^e–g^	84.16 ^a–c^	7.24 ^b,c^
Level of Significance		**	**	**	*

Means with a different letter are significantly different at the *p* > 0.01 level of significance. Note: ** indicates significant at 1% level and * indicates significant at 5% level. Here, WSG = Weed seed germination.

**Table 3 plants-10-02017-t003:** The allelopathic influence of rice accessions on percent mean germination time of barnyard grass employing the sandwich and relay seeding method.

Variety	Code	Sandwich Method		Relay Seeding Method	
		MGT (No. day^−1^)	% Reduction for MGT	MGT (No. day^−1^)	% Reduction for MGT
Control	V_0_	5.63 ^a^	0	5.81 ^a^	0
BRRIdhan 45	V_1_	4.97 ^b,c^	11.73 ^c–e^	5.19 b	10.58 ^d^
BRRIdhan 48	V_2_	5.21 ^a,b^	7.463 ^d–e^	4.95 ^b,c^	12.92 ^b–d^
Line (L50-38-8)	V_3_	4.61 ^c,d,e^	18.11 ^b,c^	4.96 ^b,c^	12.60 ^c,d^
Saitta	V_4_	4.87 ^b,c,d^	13.48 ^c–e^	5.10 ^b,c^	10.23 ^d^
BRRIdhan 75	V_5_	4.25 ^e^	24.54 ^a,b^	4.89 ^b–d^	13.61 ^b–d^
BRRIdhan 69	V_6_	4.64 ^c,d,e^	17.57 ^b–d^	4.94 ^b–d^	13.113 ^b–d^
BR17	V_7_	3.76 ^f^	33.07 ^a^	3.82 ^f^	29.12 ^a^
Line (16-3-38-9)	V_8_	4.67 ^c,d,e^	16.86 ^b–e^	4.41 ^d,e^	22.16 ^a–c^
BRRIdhan 46	V_9_	4.87 ^b,c,d^	13.43 ^c–e^	4.81 ^b–d^	15.33 ^b–d^
Putra 1	V_10_	4.72 ^c,d^	16.10 ^b–e^	4.69 ^b–e^	17.46 ^b–d^
BRRIdhan 82	V_11_	5.23 ^a,b^	6.99 ^e^	4.81 ^b–d^	15.33 ^b–d^
MRIA1	V_12_	4.91 ^b,c,d^	12.70 ^c–e^	5.04 ^b,c^	11.16 ^c,d^
BR26	V_13_	4.44 ^d,e^	21.09 ^b,c^	4.89 ^b–d^	13.88 ^b–d^
BR23	V_14_	4.24 ^e^	24.74 ^a,b^	4.31 ^e,f^	23.97 ^a,b^
MRQ74	V_15_	4.94 ^b,c^	12.42 ^c–e^	5.14 ^b,c^	9.51 ^d^
BRRIdhan 39	V_16_	4.62 ^c,d,e^	17.78 ^b–d^	4.64 ^c–e^	18.47 ^b–d^
Putra 2	V_17_	5.24 ^a,b^	6.84 ^e^	4.98 ^b,c^	12.24 ^c,d^
Level of Significance		**	**	**	*

Means with a different letter are significantly different. Here, ** indicates significance at a 1% level of significance and * at a 5% level of significance.

**Table 4 plants-10-02017-t004:** The allelopathic impact of rice accessions on percent root length of barnyard grass employing the sandwich and relay seeding method.

Variety	Code	Sandwich Method		Relay Seeding Method	
		Root Length	% Reduction of Root Length	Root Length	% Reduction of Root Length
Control	V_0_	1.59 ^a^	0	3.26 ^a^	0
BRRIdhan 45	V_1_	1.20 ^b,c^	26.42 ^e,f^	2.18 ^b–d^	32.71 ^a,b^
BRRIdhan 48	V_2_	1.48 ^a,b^	16.07 ^f^	2.19 ^b–d^	31.73 ^a,b^
Line (L50-38-8)	V_3_	1.01 ^b–d^	35.93 ^d,e^	2.07 ^b–d^	36.50 ^a,b^
Saitta	V_4_	0.83 ^d–g^	47.39 ^a–d^	3.05 ^a^	6.57 ^c^
BRRIdhan 75	V_5_	0.95 ^c–f^	40.12 ^b–e^	2.19 ^b–d^	32.37 ^a,b^
BRRIdhan 69	V_6_	0.99 ^c–e^	37.67 ^c–e^	2.53 ^b^	21.61 ^b,c^
BR17	V_7_	0.66 ^g^	58.15 ^a^	1.71 ^d^	47.58 ^a^
Line (16-3-38-9)	V_8_	0.79 ^d–g^	50.00 ^a–d^	1.87 ^c,d^	42.31 ^a^
BRRIdhan 46	V_9_	0.75 ^e–g^	52.31 ^a–c^	2.04 ^b–d^	37.29 ^a^
Putra 1	V_10_	0.84 ^d–g^	47.12 ^a–d^	2.03 ^b–d^	37.50 ^a,b^
BRRIdhan 82	V_11_	0.90 ^d–g^	43.25 ^a–d^	2.34 ^b,c^	29.20 ^a,b^
MRIA1	V_12_	1.15 ^b,c^	27.24 ^e,f^	1.98 ^b–d^	38.81 ^a,b^
BR26	V_13_	0.79 ^d–g^	50.27 ^a–d^	1.86 ^c,d^	43.06 ^a^
BR23	V_14_	0.71 ^f,g^	55.18 ^a,b^	1.77 ^c,d^	45.65 ^a^
MRQ74	V_15_	0.81 ^d–g^	49.41 ^a–d^	1.82 ^c,d^	44.33 ^a^
BRRIdhan 39	V_16_	0.78 ^d–g^	51.09 ^a–d^	2.13 ^b–d^	34.44 ^a,b^
Putra 2	V_17_	0.83 ^d–g^	47.59 ^a–d^	2.23 ^b–d^	31.255 ^a,b^
Level of Significance		**	**	**	*

Means with a different letter are significantly different. Here, ** indicates significant at a 1% level significance and * indicates significant at a 5% level of significance.

**Table 5 plants-10-02017-t005:** The allelopathic effect of rice accessions on percent shoot length of barnyard grass employing the sandwich and relay seeding method.

Variety	Code	Sandwich Method		Relay Seeding Method	
		Shoot Length	% Reduction for Shoot Length	Shoot Length	% Reduction for Shoot Length
Control	V_0_	3.4 ^a^	0	3.28 ^a^	0
BRRIdhan 45	V_1_	2.59 ^b,c^	23.72 ^c,d^	2.22 ^b^	31.16 ^a,b^
BRRIdhan 48	V_2_	3.06 ^a,b^	9.96 ^d^	2.48 ^a,b^	24.28 ^a,b^
Line (L50-38-8)	V_3_	2.17 ^c,d^	35.38 ^a–c^	2.50 ^a,b^	23.15 ^a,b^
Saitta	V_4_	2.05 ^c,d^	38.99 ^a–c^	2.67 ^a,b^	16.99 ^b^
BRRIdhan 75	V_5_	2.04 ^b–d^	39.91 ^a–c^	2.46 ^a,b^	24.23 ^a,b^
BRRIdhan 69	V_6_	1.83 ^d^	46.11 ^a,b^	2.42 ^a,b^	25.91 ^a,b^
BR17	V_7_	1.64 ^d^	51.90 ^a^	2.02 ^b^	36.75 ^a^
Line (16-3-38-9)	V_8_	2.15 ^c,d^	36.82 ^a–c^	2.47 ^a,b^	25.21 ^a,b^
BRRIdhan 46	V_9_	2.11 ^c,d^	37.68 ^a–c^	2.12 ^b^	34.96 ^a,b^
Putra 1	V_10_	1.85 ^d^	45.25 ^a,b^	2.48 ^a,b^	24.63 ^a,b^
BRRIdhan 82	V_11_	2.75 ^b^	19.11 ^c,d^	2.27 ^a,b^	30.09 ^a,b^
MRIA1	V_12_	2.54 ^b,c^	25.75 ^b–d^	2.36 ^a,b^	26.77 ^a,b^
BR26	V_13_	1.75 ^d^	48.53 ^a^	1.94 ^b^	40.61 a
BR23	V_14_	1.70 ^d^	49.92 ^a^	2.15 ^a,b^	32.48 ^a,b^
MRQ74	V_15_	2.07 ^c,d^	38.65 ^a–c^	2.38 ^a,b^	27.86 ^a,b^
BRRIdhan 39	V_16_	1.75 ^d^	48.30 ^a^	2.37 ^a,b^	27.01 ^a,b^
Putra 2	V_17_	2.05 ^c,d^	39.39 ^a–c^	2.52 ^a,b^	26.15 ^a,b^
Level of Significance		**	**	NS	NS

Means with a different letter are significantly different. Here, ** indicates significant at a 1% level significance and NS means non-significant.

**Table 6 plants-10-02017-t006:** The allelopathic influence of rice accessions on percent dry matter of barnyard grass employing the sandwich and relay seeding method.

Variety	Code	Sandwich Method		Relay Seeding Method	
		Dry Matter (mg)	% Reduction of Dry Matter	Dry Matter (mg)	% Reduction of Dry Matter
Control	V_0_	7.5 ^a^	0	43 ^a^	0
BRRIdhan 45	V_1_	5.75 ^b–d^	22.77 ^a–c^	16 ^e–g^	62.36 ^b–d^
BRRIdhan 48	V_2_	6.75 ^a–c^	9.38 ^b,c^	16 ^e–g^	62.60 ^b–d^
Line (L50-38-8)	V_3_	6.5 ^a–c^	13.84 ^b,c^	18 ^d–f^	57.66 ^c,d^
Saitta	V_4_	7.25 ^a,b^	3.23 c	19.2 ^c,d^	54.65 ^d–f^
BRRIdhan 75	V_5_	6.75 ^a–c^	13.4 ^b,c^	15 ^e–h^	65.10 ^a–d^
BRRIdhan 69	V_6_	6.75 ^a–c^	16.52 ^a–c^	23 ^b,c^	46.11 ^f–g^
BR17	V_7_	4.75 ^d^	36.16 ^a^	10.5 ^i^	75.23 ^a^
Line (16-3-38-9)	V_8_	5.50 ^d,c^	25.90 ^a,b^	13 ^g–i^	69.29 ^a,b^
BRRIdhan 46	V_9_	6.25 ^a–c^	16.07 ^a–c^	11 ^h,i^	74.23 ^a^
Putra 1	V_10_	5.75 ^b–d^	23.66 ^a–c^	19 ^c,d^	55.513 ^d–f^
BRRIdhan 82	V_11_	6.00 ^a–d^	19.64 ^a–d^	21 ^b–d^	51.02 ^e–g^
MRIA1	V_12_	7.35 ^a,b^	3.13 ^b–e^	18.5 ^d,e^	56.38 ^d–f^
BR26	V_13_	6.00 ^a–d^	19.64 ^a–c^	15 ^e–h^	64.86 ^a–d^
BR23	V_14_	6.00 ^a–d^	19.64 ^a–c^	14.5 ^i^	75.35 ^a^
MRQ74	V_15_	7.00 ^a–c^	6.25 ^b,c^	24 b	43.55 ^g^
BRRIdhan 39	V_16_	6.25 ^a–c^	16.07 ^a–c^	14 ^f–i^	67.30 ^a–c^
Putra 2	V_17_	7.00 ^a–c^	6.25 ^b,c^	19 ^c,d^	55.48 ^d–f^
Level of Significance		**	*	**	**

Means with a different letter are significantly different. Here, ** indicates significant at a 1% level significance and * indicates significant at a 5% level of significance.

**Table 7 plants-10-02017-t007:** The allelopathic consequence of rice accessions on average percent inhibition of barnyard grass employing the sandwich and relay seeding method.

Variety	Code	Sandwich Method	Relay Seeding Method
		API	API
BRRIdhan 45	V_1_	18.21 ^g,h^	28.63 ^c–e^
BRRIdhan 48	V_2_	5.81 ^i^	28.14 ^c–e^
Line (L50-38-8)	V_3_	23.15 ^d–g^	27.80 ^c–e^
Saitta	V_4_	21.18 ^e–h^	18.79 ^f^
BRRIdhan 75	V_5_	27.43 ^b–d^	28.31 ^c–e^
BRRIdhan 69	V_6_	21.04 ^e–h^	23.00 ^e,f^
BR17	V_7_	40.62 ^a^	41.52 ^a^
Line (16-3-38-9)	V_8_	29.75 ^b–d^	34.41 ^a–c^
BRRIdhan 46	V_9_	25.23 ^e,f^	34.00 ^a–d^
Putra 1	V_10_	28.93 ^b–d^	29.405 ^c–e^
BRRIdhan 82	V_11_	18.53 ^f–h^	26.4 ^c–e^
MRIA1	V_12_	15.43 ^h^	27.873 ^c–e^
BR26	V_13_	31.07 ^b,c^	34.293 ^a–c^
BR23	V_14_	34.14 ^a,b^	38.553 ^a,b^
MRQ74	V_15_	21.75 ^e–h^	26.143 ^d,e^
BRRIdhan 39	V_16_	28.92 ^b–d^	32.218 ^b–d^
Putra 2	V_17_	21.35 ^e–h^	26.47 ^c–e^
Level of significance	-	**	**

Means with a different letter are significantly different at *p* > 0.01 level of significance. ** indicates a 1% level of significance. Here, API = average percent inhibition.

**Table 8 plants-10-02017-t008:** Pearson’s correlation coefficient among the six quantitative traits of 17 rice accessions.

	M	Ger (%)	RL	SL	DW	MGT	API
Ger (%)	R	1					
	S	1					
RL	R	0.26 *	1				
	S	0.38 *	1				
SL	R	0.13 *	0.36 *	1			
	S	0.58 **	0.49 **	1			
DW	R	0.34 *	0.26 *	0.30 *	1		
	S	0.29 *	0.18 *	0.07 *	1		
MGT	R	0.89 **	0.30 *	0.14 *	0.40 *	1	
	S	0.81 **	0.30 *	0.51 **	0.31 *	1	
API	R	0.64 **	0.73 **	0.62 **	0.68 **	0.69 **	1
	S	0.78 **	0.70 **	0.77 **	0.54 **	0.74 **	1

Note: Here, ** indicates significant at a 1% level significance and * indicates significant at a 5% level of significance. M = Methods, Ger = Germination, MGT = Mean Germination Time, RL = Root Length, SL = Shoot Length, DW = Dry Weight, API = Average Percent Inhibition R = Relay seeding, S = Sandwich.

**Table 9 plants-10-02017-t009:** Cluster analysis showing the similarity group among the rice accessions.

Cluster	No. of Accessions	Accessions	Origin
I	V_1_, V_2_, V_7_, V_8_, V_10_,	BRRIdhan 45, BRRIdhan 48, BR17, Line (16-3-38-9)	BRRI, Bangladesh, UPM, Malaysia
II	V_3_, V_5_, V_13_, V_14_, V_16_	Line (L50-38-8), BRRIdhan 75, BR26, BR23, BRRIdhan 39	UPM, Malaysia, BRRI, Bangladesh,
III	V_4_, V_9_, V_15_, V_17_	Saitta, BRRIdhan 46, MRQ74, Putra2	BRRI, Bangladesh, MARDI, Malaysia, UPM, Malaysia
IV	V_12_	MRIA1	MARDI, Malaysia
V	V_6_	BRRIdhan69	BRRI, Bangladesh
VI	V_11_	BRRIdhan82	BRRI, Bangladesh

## Data Availability

Not applicable.

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
