# Peer review of "Allelopathic Effect of Selected Rice (Oryza sativa) Varieties against Barnyard Grass (Echinochloa cruss-gulli)"

_plants, 2021, doi:10.3390/plants10102017_

Round 1
Reviewer 1 Report
Dear Authors,
The manuscript presents a study about selection of potential allelopathic rice varieties against barnyard grass (Echinochloa crus-galli), one of the most troublesome weeds in rice cropping systems all over the world. Allelopathy is an interesting approach for reduction chemical weed control in both organic as well middle term agroecosystems, such as IWM.
However, nowadays allelopathic studies to be published requires field or semi-field trials, as bioassays results can not be directly extrapolated to field conditions.
The biochemical interaction among plants is most regulated by biotic factors. The buffer action of soil is one of the main factors affecting allelopathy at field level where microorganisms interaction plays a significant role. Also weed density, particularly of E.crus-galli regulates the response of the crop, hence the allelopathic potential of each rice variety.
I would like also to point out two or three specific comments:
Keywords - select words that do not repeat the ones in the title
In M&M it is not clear which plant part is responsible for allelopathic effects. Each method studies the effects of different plant parts.
In R&D Please confirm whether two known alleopathic varieties (V7 and V8) were used as standard references to confirm alleolopathy.
I propose to reject the manuscript in the present format, because aditional experiments are needed.
Reviewer 2 Report
The manuscript "Allelopathic effect of Selected Rice (Oryza sativa) Varieties against Barnyard Grass (Echinochloa cruss-gulli)" Its local interest But the findings are significant.
- The species must be in Italics (see line 35, 36 etc)
- line 97- 99 . ....environment friendly ... Too general comment. Please rephrase.
- line 148 Petridis or Petri? Please define
- line 184-185, API . I THINK ITS better to write as equation
- RESULTS & DISCUSSION. The authors must to separate that to different parts.
- The Figure 1 Its better to replaced with Table
- Figure 2 and Table 7 show the same. I suggest to keep the table
- Figure 3 and 4 are in low quality (in printed form) Please increase the analysis
- Ref 26 Its in Arabic . I suggest to translate in English
Reviewer 3 Report
The research is devoted to the comparative study by two bioassays of 17 selected rice lines that have allelopathic potential to suppress germination patterns and biometric traits of weed barnyard grass (Echinochloa cruss-gulli). The paper presents an interesting result that has fundamental and practical meaning.
However, I have some comments and observations:
Introduction: is well written, well documented with relevant literature and it is able to introduce and familiarize with the subject of the study.
Materials and Methods: clear, logical and easily replicable. However formula (1) between lines 171-172 seems incorrect. Check it again. “Sown” seeds on Petri dish covered with filter paper?!? – It is more appropriate to use “put”. (line 152)
Results are presented in 5 tables and 4 figures in the main body along with Discussion part. I think it is better to separate the results from the discussion. In addition, the results for each indicator should be separated in a subheading and with the corresponding table, comparing the two methods. For example, Table 2 should present only the results for Weed seed germination determined by both methods (from Table 2 and Table 4), together with the reduction percentage for this parameter from Table 3 and Table 5. Similarly, Table 3 should present Mean Germination Time; Table 4 - Root length; Table 5 - shoot length; Table 6 - dry weight; and Table 7 - only Average Percent Inhibition data calculated for both bioassayс. This will make the data easier to perceive. Furthermore cluster analyses presented on 3 figures (Figures 2-4) and one Table (Table 7) do not give any additional information. In essence, the data are the same, they just look like different and I think it's completely unnecessary. You can present only one type of figure of your choice. I would recommend Figure 3.
Other observation:
- “…and it was near to the BRRIdhan 45 (1.69%).” (lines 207-208) seems incorrect.
- Remove lines 224-228, which seem to be a part of Results & Discussion, from explanations under Table 2.
- Omit g from 0.66 g (line 248)
- Revise sentences on lines 250-251 and lines 287-289.
- line 378 – 2.02 mm is not lowest position. Revise the sentence.
- line 396 – change “..BR17 (29.12%)…” with BR17 (75.23%).
- lines 359 and 391 – “…smashed rice seeds…” should be removed. In this bioassay you used intact rice seedlings!
- lines 420-421 “In the Relay method, the API ranged between 42.93% and 19.52%.“– I did not see such values in the Table 5.
Conclusions: concise and very well presented.
References: the cited references correlated well with the text
Reviewer 4 Report
In the manuscript entitled: "Allelopathic effect of Selected Rice (Oryza sativa) Varieties against Barnyard Grass (Echinochloa cruss-gulli)." (number: plants-1302051), the Authors described an interesting topic concerning the industrially important plant in the context of sustainable agriculture. Selecting rice for allelopathy is an excellent natural technique to increase the yield of this extremely important plant for humans.
In general, in my opinion, the work is well written, and contain interesting data for broad audience. However, there are some inaccuracies, which are listed below.
Page 1, line 35: “Oryza sativa” – it should be Italic.
Pge 1, line 43: “Weeds are unwanted plants that are of no economic value…” – in my opinion, the so-called "weeds" are difficult to assess economically, as they are extremely important for the biodiversity of the environment. It is probably better to say that they are “lowering crops production”, because I am afraid that in the overall economic balance of the Earth, the presence of "weeds" will turn out to be crucial.
Page 2, line 87: “…in some rice accessions from Bangladesh…” – it is not clear word “accession”. I think it should be “cultivars” – please clarify your intention.
In general, in my opinion the presented study describes the allelopathic correlation between two plant species, and despite the fact that the research concerned only a few traits (seed germination and seedling growth), a very good statistical analysis of the results allowed the authors to draw important conclusions.
Round 2
Reviewer 2 Report
The authors cover all comments
Author Response
Dear sir
Already revised as per comments and suggestions. Thanks.